# Carbon Felts Uniformly Modified with Bismuth Nanoparticles for Efficient Vanadium Redox Flow Batteries

**DOI:** 10.3390/nano14242055

**Published:** 2024-12-23

**Authors:** Huishan Chen, Sen Li, Yongxin Zhao, Xinyue Li, Hui Zhao, Longzhen Cheng, Renting Li, Pengcheng Dai

**Affiliations:** State Key Laboratory of Heavy Oil Processing, Institute of New Energy, College of New Energy, China University of Petroleum (East China), Qingdao 266580, China; s23030052@s.upc.edu.cn (H.C.); 15610533203m@sina.cn (S.L.); s22030079@s.upc.edu.cn (Y.Z.); z22030111@s.upc.edu.cn (X.L.); z23030097@s.upc.edu.cn (H.Z.); z23030063@s.upc.edu.cn (L.C.); li15550819583@sina.com (R.L.)

**Keywords:** bismuth nanoparticles, electrode modification, vanadium redox flow batteries, catalytic activity

## Abstract

The integration of intermittent renewable energy sources into the energy supply has driven the need for large-scale energy storage technologies. Vanadium redox flow batteries (VRFBs) are considered promising due to their long lifespan, high safety, and flexible design. However, the graphite felt (GF) electrode, a critical component of VRFBs, faces challenges due to the scarcity of active sites, leading to low electrochemical activity. Herein, we developed a bismuth nanoparticle uniformly modified graphite felt (Bi-GF) electrode using a bismuth oxide-mediated hydrothermal pyrolysis method. The Bi-GF electrode demonstrated significantly improved electrochemical performance, with higher peak current densities and lower charge transfer resistance than those of the pristine GF. VRFBs utilizing Bi-GF electrodes achieved a charge-discharge capacity exceeding 700 mAh at 200 mA/cm^2^, with a voltage efficiency above 84%, an energy efficiency of 83.05%, and an electrolyte utilization rate exceeding 70%. This work provides new insights into the design and development of efficient electrodes, which is of great significance for improving the efficiency and reducing the cost of VRFBs.

## 1. Introduction

The rapid integration of intermittent renewable energy sources, such as wind and solar power, into energy supply has necessitated the development of large-scale energy storage technologies [1,2,3]. Vanadium redox flow batteries (VRFBs), which utilize vanadium ions in both the positive and negative electrodes as active materials, have garnered significant attention due to their long lifespan, high safety, and flexible structural design [4,5]. However, the relatively low power and energy density of VRFBs lead to higher system costs ($135–210 per kWh) [6,7,8], which hinders their widespread adoption.

The electrode is the core crucial component of VRFBs, as it determines the reaction rate of the vanadium ions in essence [9,10]. Graphite felt (GF) currently serves as the primary electrode material due to its porous structure, excellent chemical stability, and good electrical conductivity [11,12]. However, the scarcity of active sites leads to low electrochemical activity, which is the predominant challenge faced by GF, limiting the overall performance of VRFBs [13,14]. Surface functionalization of GF with oxygen- or nitrogen-containing functional groups has been proven to introduce more active sites that facilitate the redox of vanadium species. Nevertheless, these functional groups usually facilitate the hydrogen evolution reaction (HER) [15], leading to irreversible changes in the valence state of vanadium ions, which results in rapid deterioration of the electrolyte [16]. Suárez, D.J. group reported that modifying GFs with bismuth nanoparticles (Bi NPs) facilitates the formation of the intermediate product BiH_x_, which is capable of enhancing the redox reversibility of V^3+^ to V^2+^ and thus suppresses the competing HER. However, the methods for introducing Bi NPs onto GF are primarily electrochemical deposition and thermal reduction, which can hardly achieve a uniform distribution of bismuth nanoparticles on the three-dimensional scale of the carbon felt [17,18]. Therefore, developing a new strategy that can uniformly and firmly decorate Bi NPs on the surface of GF is crucial for improving the performance of VRFBs.

In this work, we employed a solvothermal strategy to deposit bismuth oxide on the surface of carbon felt uniformly and subsequently achieved a uniform distribution of bismuth nanoparticles on the surface of the carbon felt electrode (Bi-GF) through in situ carbothermal reduction. Compared with the GF electrode, the Bi-GF electrode shows higher peak current densities and lower charge transfer resistance on both the positive and negative sides. Bi-GF electrode exhibits superior electrochemical catalytic performance, with a charge-discharge capacity exceeding 700 mAh at a current density of 200 mA/cm^2^, which is higher than that of 200 mAh for the heat-treated GF. Furthermore, the voltage efficiency (VE) of the Bi-GF electrode was greater than 84%, with an EE of 83.05% and an electrolyte utilization rate (EU) exceeding 70% at a current density of 200 mA/cm^2^. This research offers novel perspectives on the creation and enhancement of effective electrodes, which hold substantial implications for enhancing the efficiency and reducing the cost of VRFBs.

## 2. Materials and Methods

### 2.1. Materials and Chemicals

Graphite felt (G280A, AvCarb, Wiesbaden, Germany) was purchased from the SGL Carbon Company (DE). Bismuth sulfate (Bi_2_(SO_4_)_3_, 99.9%) and vanadyl sulfate (VOSO_4_, 99.9%) were obtained from Shanghai Macklin Biochemical Technology Co., Ltd. (Shanghai, China). Nitric acid (HNO_3_, 68%) and concentrated sulfuric acid (H_2_SO_4_, 98%) were purchased from Sinopharm Chemical Reagent Co., Ltd. (Shanghai, China). Argon (Ar, 99.99%) was purchased from Tianyuan Gas Co., Ltd. (Qingdao, China), and deionized water was prepared in the laboratory.

### 2.2. Preparation of Bi-GF

Graphite felt in the size of 2.7 cm × 2.7 cm was washed with deionized water several times and followed by dried at 80 °C. Pristine graphite felt (P-GF) was calcined at 500 °C for 5 h to obtain the thermally treated graphite felt serving as a comparative sample. Bi_2_(SO_4_)_3_ (1.07 g) was dissolved in 30 mL of 2 M diluted HNO_3_ and ultrasonically stirred for 1 h. Then, the GF was immersed in the mixed solution, hydrothermally treated at 180 °C for 5 h, and finally washed with deionized water to obtain an intermediate Bi_2_O_x_-GF. After drying at 80 °C, Bi_2_O_x_-GF was calcined at 200 °C for 1 h at a heating rate of 5 °C/min with Ar flowing, and then the temperature was raised to 700 °C at a heating rate of 5 °C/min, which reduced the Bi_2_O_x_ nanoparticles to elemental Bi. Finally, the obtained product Bi-GF was washed with deionized water and then dried at 80 °C.

### 2.3. Physicochemical Characterization

A field emission scanning electron microscope (SEM, JEM-7500F, JEOL, Musashino, Tokyo) was employed to examine the surface morphology of the materials. The mass loading of Bi was determined by weighing GF and Bi-GF and calculating the difference in mass per unit area. A transmission electron microscope (TEM, Tecnai G2F20, FEI, Hillsboro, OR, USA) was used to study the internal structure and size distribution of nanoparticles. Energy dispersive X-ray spectroscopy (EDS) was used for chemical characterization and elemental analysis of the materials. An X-ray diffractometer (XRD, χPert Pro, Panaco, The Netherlands) was used to analyze the composition and structure of the materials and the crystalline phase. The scattering intensity is a function of the incident angle, scattering angle, polarization, and wavelength or energy. A Raman spectrometer (Raman, HR Evolution, HORIBA Scientific, Edison, NJ, USA) was employed to probe the graphitization degree of the materials. A surfaspector (SDC-350Z, Sindin, Dongguan, China) was used to analyze the contact angle. X-ray photoelectron spectroscopy (XPS, ESCALAB 250Xi, ThermoFisher, Waltham, MA, USA) was used to investigate the chemical composition and valence states. Nitrogen adsorption-desorption isotherms (Autosorb-iQ2, Quantachrome, Boynton Beach, FL, USA) were used for the specific surface area and porosity analysis.

### 2.4. Electrochemical Measurements

Cyclic voltammetry (CV) and electrochemical impedance spectroscopy (EIS) were performed with a typical three-electrode system using a CHI 760E electrochemical workstation (Shanghai Chenhua Instrument Co., Ltd, Shanghai, China) at room temperature. The three-electrode system was fabricated with a GF or Bi-GF working electrode (area of 2.7 cm × 2.7 cm), a platinum plate counter electrode (area of 1 × 1 cm), and a saturated calomel reference electrode. The electrolyte used for the positive electrode was a 20 mL solution containing 0.1 M VO^2+^ and 3 M H_2_SO_4_, prepared by dissolving 2 mmol VOSO_4_ in 20 mL 3 M H_2_SO_4_. The electrolyte used for the negative electrode was a 20 mL solution containing 0.1 M V^3+^ and 3 M H_2_SO_4_, obtained by dissolving 1 mmol V_2_(SO_4_)_3_ in 20 mL 3 M H_2_SO_4_. Cyclic voltammetry (CV) was carried out at different scan rates (from 2 mV/s to 20 mV/s), and the voltage ranges of CV tests for positive reaction and negative reaction were 0.35 to 1.45 V (vs. Hg/Hg_2_Cl_2_) and −1.1 to 0 V (vs. Hg/Hg_2_Cl_2_), respectively. EIS was conducted in the frequency range of 10^−1^–10^5^ Hz, and the polarization potentials were set as the open-circuit potentials, which were 0.75 V (vs. Hg/Hg_2_Cl_2_) and −0.38 V (vs. Hg/Hg_2_Cl_2_) for positive and negative reactions, respectively. The electrochemically active surface area (EASA) was estimated by measuring the double-layer capacitance (C_dl_) of the system using CV. The equations are as follows (Equations (1) and (2)) [19,20]:(1)$$i_{c}=vC_{dl}$$
where i_c_ is the charging current and v is the scan rate.
(2)$$EASA=\frac{C_{dl}}{C_{S}}$$
where C_dl_ is the double-layer capacitance and C_s_ is the specific capacitance.

### 2.5. VRFB Performance Test

The VRFB system consists of two polytetrafluoroethylene backplates and floating frames, two electrodes, an ion exchange membrane (Nafion 212 membrane), two graphite bipolar plates, two copper current collectors, and peristaltic pumps. The electrolyte used for the positive and negative electrodes was a 20 mL solution containing 1.7 M VO^2+^ and 3 M H_2_SO_4_, 20 mL solution containing 1.7 M V^3+^ and 3 M H_2_SO_4_, respectively. Firstly, the positive and negative electrolytes were both solutions containing 1.7 M VO^2+^ and 3 M H_2_SO_4_ (purchased from Wuhan Zhisheng New Energy Co., Ltd., Wuhan, China). To obtain the negative electrolyte, a pre-electrolysis process is required to convert VO^2+^ to V^3+^ via constant-current charging at 0.45 A for 50 min. Then, the VO_2_^+^ electrolyte on the positive side was replaced with an equal amount of fresh VO^2+^ electrolyte. Both compartments received a continuous flow of electrolytes at a rate of 60 mL/min. The single-cell performance test was performed at various current densities ranging from 200 to 450 mA/cm^2^ and a potential window between 1.65 and 0.8 V. Voltage loss refers to the change in voltage across an electrical component. It is defined as the difference between the open-circuit voltage and the initial charging voltage [21]. In the process of operation, reversible redox reactions occur at the interface between the electrodes and the electrolyte. The reaction equations are shown as follows (Equations (3)–(8)) [22,23]:

On the positive side:(3)$$\mathrm{charge}:\;{\mathrm{VO}}^{2+}+H_{2}O\;\to \;VO_{2}^{+}\;+\;2H^{+}\;+\;e^{-}$$
(4)$$\mathrm{discharge}:\;{{\mathrm{VO}}_{2}}^{+}+2H^{+}+e^{-}\;\to \;{\mathrm{VO}}^{2+}\;+\;H_{2}O$$

On the negative side:(5)$$\mathrm{charge}:\;V^{3+}+e^{-}\;\to \;V^{2+}$$
(6)$$\mathrm{discharge}:\;V^{2+}\;\to \;V^{3+}\;+\;e^{-}$$

The overall battery reaction is as follows:(7)$$\mathrm{charge}:\;{VO}^{2+}+V^{3+}+H_{2}O\;\to \;{VO}_{2}^{+}\;+\;V^{2+}\;+\;2H^{+}$$
(8)$$\mathrm{discharge}:\;{\mathrm{VO}}_{2}^{+}+V^{2+}+2H^{+}\;\to \;{\mathrm{VO}}^{2+}\;+\;V^{3+}\;+\;H_{2}O$$

## 3. Results

### 3.1. Structural and Compositional Analyses

The synthetic strategy for Bi-GF is illustrated in Figure 1a. P-GF exhibits a smooth surface with no substances attached (Appendix A). After the hydrothermal process, many irregular nanoparticles were decorated on the surface of GF (Appendix A). The XRD pattern (Appendix A) reveals that these irregular nanoparticles are bismuth oxides (Bi_2_O_3_ and Bi_2_O_2.3_). The bismuth oxides were then reduced to Bi NPs that were decorated uniformly and firmly on the surface of GF via an in situ carbothermal reduction process [24] (Figure 1b,c). By calculating the difference in mass per unit area of GF and Bi-GF, it was revealed that the mass loading of Bi was 0.66 mg/cm^2^. As illustrated in Figure 1b, the high-resolution TEM image shows the lattice fringe spacings of 0.257 nm and 0.342 nm correspond to the (0 −1 1) and (1 0 1) lattice planes of elemental Bi (Figure 1d). The elemental mapping further confirms that the Bi nanoparticles are uniformly distributed on the surface of GF, as illustrated in Figure 1e. As depicted in Figure 1f, the XRD pattern of Bi-GF shows two types of diffraction peaks corresponding to the elemental Bi and C, respectively. These results above demonstrate the uniform distribution of Bi NPs. in Bi-GF. Figure 1g shows the Raman spectrum of Bi-GF. Two significant peaks at 1356 cm^−1^ and 1596 cm^−1^ are attributed to the D and G peaks, respectively [25]. The I_D_/I_G_ intensity ratio of Bi-GF was calculated to be 1.37, higher than that of 1.26 for GF, suggesting that Bi-GF possesses abundant defects on the surface during the Bi NPs decoration process [26]. As shown in Appendix A, the pore diameter distributions of Bi-GF and GF range from 4 to 25 nm and are mostly concentrated at 6 nm. The Brunauer–Emmet–Teller (BET) specific surface area of Bi-GF is 5.862 m^2^/g, which is almost ten times that of GF (0.624 m^2^/g) (Appendix A).

The XPS survey spectrum of Bi-GF is shown in Figure 2a, which reveals the presence of C, Bi, and O elements. The C 1s XPS spectrum (Figure 2b) displays the peaks located at 283.6 eV, which correspond to the C-Bi bonds [27]. Figure 2c shows the Bi 4f XPS spectrum, in which the peaks at 159.7 eV and 165.1 eV are assigned to metallic Bi^0^ [28], and the peaks at 160.98 eV and 166.08 eV coincide with the C-Bi bonds [29]. The existence of C-Bi bonds can be attributed to the reaction of bismuth oxides and GF during the carbothermic process, which confirms that the Bi NPs are firmly connected to the GF surface. The contact angle measurements were performed to analyze the hydrophilicity of the electrodes. Appendix A shows that the contact angle of Bi-GF is 137.5°, which is lower than that of P-GF (141.3°), suggesting that Bi-GF possesses better hydrophilicity, benefiting electrolyte accessibility [30].

### 3.2. Electrocatalytic Activities of Bi-GF

To investigate the electrochemical performance, the CV curves of GF and Bi-GF as positive electrodes were obtained. As shown in Figure 3a, the oxidation peak current (I_pa_) and reduction peak current (I_pc_) of Bi-GF are higher than those of GF, suggesting that Bi-GF possesses excellent redox catalytic activity for the VO^2+^/VO_2_^+^ redox couple. Moreover, in comparison with the 20-cycle CV curves of the GF electrode (Appendix A), the CV curves of the Bi-GF (Appendix A) electrode display less fluctuation and enhanced stability. Subsequently, Figure 3b demonstrates the CV curves of the negative electrode. The Bi-GF electrode exhibits an oxidation peak current density of 0.28 mA/cm^2^ and a reduction peak current density of −0.39 mA/cm^2^, larger than those of GF, suggesting that Bi-GF has superior catalytic performance for the V^3+^/V^2+^ redox reaction. Similarly, the Bi-GF electrode exhibits more stable 20-cycle CV curves (Appendix A), which reveals the stability of Bi-GF for the V^3+^/V^2+^ electrocatalytic process. In addition, as depicted in Appendix A, the redox peak currents improve with the enhancement of the scanning rate, but also lead to an increase in the peak potential difference owing to the increase in electrochemical polarization [31]. Appendix A reveals an approximately linear correlation between the peak current of the Bi-GF electrode and the square root of the scan rate, suggesting that the VO^2+^/VO_2_^+^ reaction on the Bi-GF electrode is diffusion-controlled [32]. Moreover, the −I_pc_/I_pa_ values at different scan rates are shown in Appendix A. The −I_pc_/I_pa_ ratio of Bi-GF is close to 1.0, suggesting excellent redox reversibility [33]. Notably, compared to GF, the Bi-GF exhibited a significant increase in BET-specific surface area. To determine whether the performance enhancement of Bi-GF is due to an increase in the BET-specific surface area or the introduction of Bi NPs, we conducted an analysis of the electrochemically active surface area (EASA). It is found that the EASAs of Bi-GF (0.249 cm^2^) and GF (0.234 cm^2^) are nearly identical (Appendix A), which may be due to the inability of the electrolyte to access the nanoscale pores [34]. Therefore, the improved performance of Bi-GF is attributed to the catalytic effect of Bi NPs rather than a simple increase in the BET-specific surface area.

Figure 3c,d shows the Nyquist plots of GF and Bi-GF as positive and negative electrodes, with all spectra fitted using the same equivalent circuit, as shown in the inset, where R_s_, R_ct_, Z_w,_ and CPE denote the bulk solution resistance, charge transfer resistance, Warburg impedance related to the diffusion process, and electric double-layer capacitance, respectively. In Nyquist plots, the intercept on the real axis of the curve is typically approximated as the solution resistance (R_s_), the semicircles observed in the high-frequency region correspond to the battery’s charge transfer resistance (R_ct_), and the low-frequency region is related to the diffusion process of the electrolyte [24,35]. In positive electrolytes, the Bi-GF’s R_ct_ is 19.2 mΩ, which is significantly less than that of 45.0 mΩ for GF (Appendix A). In negative electrolytes, the R_ct_ of Bi-GF is 31.9 Ω, lower than that of 32.8 Ω for GF (Appendix A). These results suggest that the modified Bi NPs provide more active sites, resulting in effective redox reactions of VO^2+^/VO_2_^+^ and V^3+^/V^2+^.

### 3.3. Performance of Bi-GF Electrode in VRFBs

The charge-discharge curves of the VFRBs using Bi-GF and GF as both positive and negative electrodes are presented in Figure 4a. Compared to the GF electrode, VRFBs utilizing Bi-GF as the working electrode demonstrate remarkable electrochemical performance, achieving a charge and discharge capacity exceeding 750 mAh, which is a significant improvement over the 200 mAh capacity observed with the GF electrode. Furthermore, Bi-GF exhibits a more stable charge-discharge voltage plateau and lower overpotential, as shown in Figure 4b. The initial potential difference between charging and discharging for Bi-GF is approximately 0.2 V, which is substantially lower than the 0.5 V recorded for the GF electrode. This reduced potential difference enhances electrolyte utilization and contributes to the stable operation of the VFRBs [36]. As depicted in Figure 4c, the charge-discharge voltage loss of the Bi-GF electrode is significantly lower than that of the GF electrode across different current densities, ranging from 200 to 350 mA/cm^2^. The voltage losses for Bi-GF remain below 100 mV, suggesting that a greater proportion of the current is effectively engaged in the electrochemical processes of the battery.

Figure 5a presents the charge-discharge capacity of the Bi-GF electrode across various current densities. It is evident that the Bi-GF electrode maintains stable charge-discharge performance at high current densities ranging from 200 to 350 mA/cm^2^, demonstrating low overpotential and high charge-discharge capacities. Notably, the Bi-GF electrode continues to function effectively even at an ultra-high current density of 450 mA/cm^2^. The voltage efficiency (VE), energy efficiency (EE), and overall utilization efficiency (EU) of vanadium flow redox batteries (VRFBs) at different current densities were also evaluated. As shown in Figure 5b, the VE of the batteries assembled with the Bi-GF electrode consistently outperforms that of the GF electrode across all tested current densities. Specifically, at a current density of 200 mA/cm^2^, the VE of the Bi-GF electrode exceeds 84% (84.88%), which is approximately 17% higher than the VE of the GF electrode (67.71%). The overall EE of the Bi-GF electrode-assembled battery is also superior to that of the GF electrode. Even at a current density of 350 mA/cm^2^, the Bi-GF electrode maintains a stable EE of 65.45%, surpassing the 59.74% recorded for the GF electrode, as illustrated in Figure 5c. Additionally, as depicted in Figure 5d, the Bi-GF electrode exhibits an enhanced EU compared to the GF electrode across varying current densities. These findings suggest that the Bi-GF electrode effectively inhibits the HER and provides a greater number of active sites, thereby significantly improving the performance of VRFBs.

Maintaining long-term cyclic stability is crucial for the performance of VRFBs. To evaluate this, we conducted 100-cycle charge-discharge experiments at a current density of 200 mA/cm^2^ using different electrodes. As illustrated in Figure 6a–c, VRFBs employing Bi-GF electrodes demonstrate a higher and more stable CE (~98%), VE (~79%), and EE (~79%) compared to those using GF electrodes (CE ~ 96%, VE ~ 57%, EE ~ 55%) throughout the 100-cycle process. After 100 cycles, the discharge capacity of the Bi-GF electrode decreases from 795.4 mAh to 422.95 mAh, whereas the GF electrode shows a dramatic decline from 202.33 mAh to nearly 0 mAh (Figure 6d). Furthermore, Figure 6e illustrates that the capacity loss of the Bi-GF electrode is significantly slower than that of the GF electrode, maintaining over 56% of its capacity after 100 cycles. Extending the assessment to a more rigorous 500-cycle charging and discharging regimen, the Bi-GF electrodes exhibited negligible losses in both VE and EE (Figure 6f), underscoring their outstanding stability. Additionally, the scanning electron microscopy (SEM) analysis presented in Appendix A of the Bi-GF electrode after 100 cycles revealed no appreciable morphological alterations compared to its pristine condition (Figure 1b), thereby reinforcing the assurance of operational reliability for these VRFBs. Finally, compared to other studies in terms of performance, VRFBs utilizing Bi-GF as working electrodes demonstrate excellent EE (Appendix A).

## 4. Discussion

In summary, we successfully achieved uniform and firm modification of Bi NPs on the surface of CF through a bismuth oxide-mediated hydrothermal pyrolysis method. The uniform distribution of Bi NPs brings about a higher surface area, better wettability, and more active sites, which induces a much-reduced charge transfer resistance. As a result, the as-prepared Bi-GF exhibits superior redox catalytic activity for both VO^2+^/VO_2_^+^ and V^3+^/V^2+^ couples compared with pristine CF. VRFBs with Bi-GF electrodes achieve a charge-discharge capacity exceeding 750 mAh at 200 mA/cm^2^ and maintain more than 96% coulombic efficiency, 79% voltage efficiency, and 77% energy efficiency over the long-term cycling test, significantly improving the performance of VRFBs.

## Figures and Tables

**Figure 1 nanomaterials-14-02055-f001:**
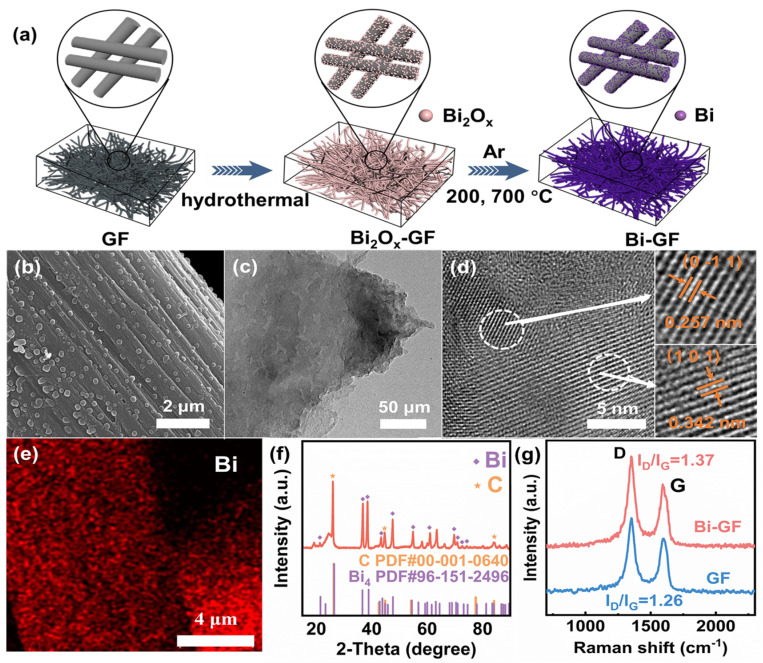
(**a**) The synthetic strategy; (**b**) SEM image; (**c**) TEM image and (**d**) HRTEM images of Bi-GF; (**e**) EDS elemental mapping; (**f**) XRD pattern of the Bi-GF; (**g**) Raman spectra of Bi-GF and GF.

**Figure 2 nanomaterials-14-02055-f002:**
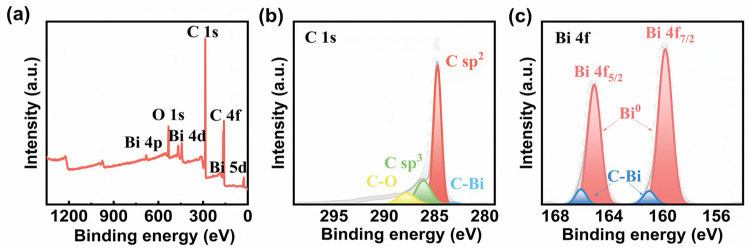
(**a**) The XPS survey spectrum; High-resolution XPS spectra of (**b**) C 1s and (**c**) Bi 4f for Bi-GF.

**Figure 3 nanomaterials-14-02055-f003:**
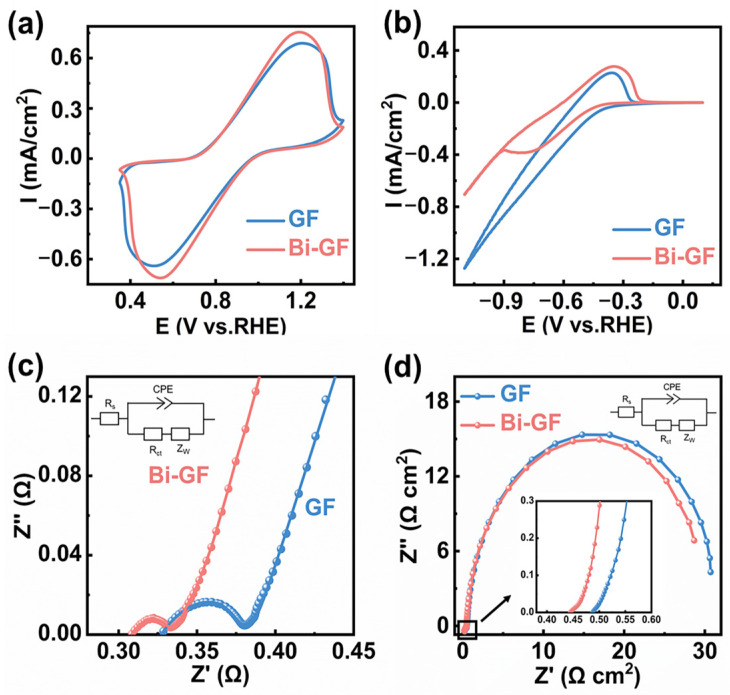
CV curves of (**a**) positive electrode of Bi-GF and GF at 5 mV/s scanning rate; (**b**) negative electrode of Bi-GF and GF at 2 mV/s scanning rate; Nyquist plots of Bi-GF and GF (**c**) as positive electrodes; and (**d**) as negative electrodes.

**Figure 4 nanomaterials-14-02055-f004:**
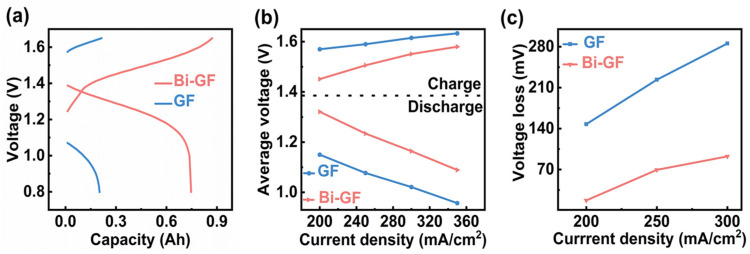
(**a**) Charge-discharge capacity curves of the Bi-GF electrode and GF electrode at a current density of 200 mA/cm^2^; (**b**) average voltage; (**c**) voltage loss.

**Figure 5 nanomaterials-14-02055-f005:**
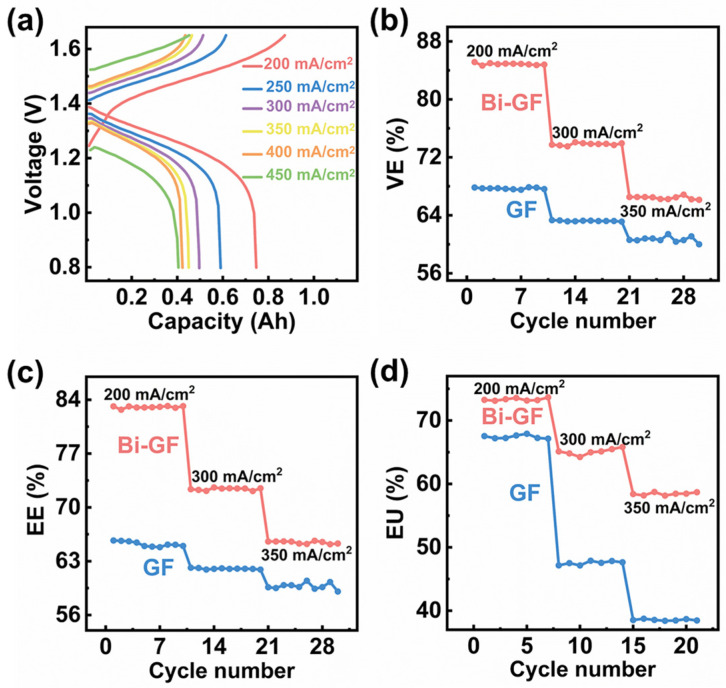
(**a**) The charge-discharge capacity curves of the Bi-GF electrode at different current densities; (**b**) voltage efficiency; (**c**) energy efficiency; (**d**) electrolyte utilization rate of Bi-GF and GF electrodes at different current densities.

**Figure 6 nanomaterials-14-02055-f006:**
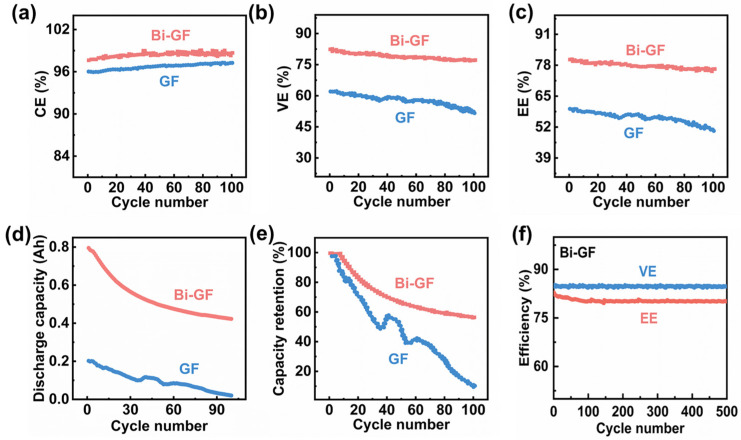
(**a**) Coulomb efficiency; (**b**) voltage efficiency (VE); (**c**) energy efficiency (EE); (**d**) discharge capacity and (**e**) capacity retention of Bi-GF and GF electrodes for 10 charge-discharge cycles; (**f**) VE and EE of the Bi-GF electrode after 500 charge-discharge cycles at 200 mA/cm^2^ current density.

## Data Availability

The data presented in this study are available on request from the corresponding author due to privacy.

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
