# Peer review of "Carbon Felts Uniformly Modified with Bismuth Nanoparticles for Efficient Vanadium Redox Flow Batteries"

_nanomaterials, 2024, doi:10.3390/nano14242055_

Round 1
Reviewer 1 Report
Comments and Suggestions for Authors
The article “Carbon felts uniformly modified with bismuth nanoparticles for efficient vanadium redox flow batteries “fits the Nanomaterials scope and is written in good English. However, the discussion of the results is not satisfactory and presents several flaws. The manuscript cannot be published in the present form.
Several issues should be addressed to improve the manuscript, some suggestions are:
Page 2, line 66-70: If you choose to present some reagents as “Bismuth sulfate (Bi2(SO4)3, 99.9%) and vanadyl sulfate (VOSO4, 99.9%)”, why do present others in a different form “Nitric acid (68%) and concentrated sulfuric acid (98%)”. You have to be consistent.
Page 2, line 67 and line 77: Please check “(Bi2(SO4)3”
Page 2, line 76: “Graphite felt” should be “graphite felt”.
Page 3, line 105: Why did the authors use Hg/Hg2Cl2 as reference electrode?
Page 3, line 106-107: Please check “0.1 M VO2+/3 M H2SO4” and “0.1 M V3+/3 M H2SO4”
Page 3, line 109: Please check “VO2+/VO2+”
Page 3, line 109-110: Do the authors performed EIS at fixed frequency “Electrochemical impedance spectroscopy (EIS) was conducted at an interference frequency of 100 kHz”?
Page 3, line 105-106: The authors said: “The electrolyte concentrations used in the two half-cells were in a 20 mL solution containing 0.1 M VO2+/3 M H2SO4 and 0.1 M V3+/3 M H2SO4, respectively”, while at page 3, line 117: “The positive and negative electrolytes are 1 M VOSO4/3 M H2SO4 solution”. Please uniformize.
Page 3, line 125: Instead of “(as shown in Equations 1 to 6)” in order to avoid repetition, it should be “(Equations 1 to 6)”
Page 3, line 126-129: Please check the reactions, they do not seem correct in terms of signs.
Supplementary materials, Table S1: The values of specific surface area should be presented with the same number of significant digits. There is a disagreement between the value of GF presented here and that in the text (page 4, line 149).
Page 4, line 162-164: The authors comment about the “hydrophilicity” of the materials, however both values achieved indicate high hydrophobicity. How can you comment on this?
Page 5, Figure 3a: The CVs presented in Figure 3a do not correspond with those in Figure S6 and S7 do not. Can you please check? Besides showing the 20 cycles stability of the electrode, the 1st scan should be the same as that in Figure 3a and is not.
Page 5, line 182-184: This statement “In addition, as depicted in Figures S10 and S11, the redox peak currents improve with the enhancement of the scanning rate, indicating that the ion diffusion rate accelerates”. The currents indeed increase with increase in scan rate, and (if represented with scan rate, or square root of scan rate) can indicate which is the process that controls the reaction. However, such plot was not realized and the diffusion rate not calculated (at least not shown) which invalidate the previous statement. Additionally, the redox behavior is compromised at higher scan rates, no comment about this was made. It is suggested to review this part of the discussion.
Page 6, line 193: The same equivalent circuit was used to fit all spectra, those in Figure 3c and 3d?
Page 6, line 193-195: The authors state “In the Nyquist plots, the semicircles observed in the low-frequency region correspond to the battery's charge transfer resistance (Rct), and those in the high-frequency region represent the diffusion impedance of the electrolyte”. Are there 2 semicircles in the plots? This is not shown. Please rewrite.
Page 5, Figure 3c: The values observed in the Figure 3c do not correspond to those discussed at page 6, line 196-198, are the units OK in the graph? Please, check.
Page 6, line 202-203: Please check the text, there is an unfinished sentence and confusing: “Compared to the GF electrode, VRFBs utilizing 202 Bi-GF as working electrodes The results indicate that VFRBs utilizing Bi-GF”.
Page 6, Figure 4c: The same number of points should be represented in the figure for GF and Bi-GF.
Page 7, line 257: The value of the capacity maintained is wrong, please check the value again.
Supplementary materials, Figure S13: The authors should present the SEM before cycling nearby to be easy to compare.
General remark: Comparison with literature in terms of performance is missing.
Reviewer 2 Report
Comments and Suggestions for Authors
The manuscript titled 'Carbon felts uniformly modified with bismuth nanoparticles for efficient vanadium redox flow batteries' presents experimental work aimed at improving the kinetics of electrochemical reactions in vanadium redox flow batteries (VRFBs) through electrode modification. While the concept of bismuth metal deposition on carbon felt electrodes for VRFBs has been explored previously, this study introduces a novel method for obtaining carbon felt modified with bismuth nanoparticles.
To evaluate the effectiveness of this new modification method, the obtained parameters for the modified electrode should be compared with similar data reported for conventional systems. Additionally, a comparative analysis with thermally activated carbon felt is essential. However, the electrochemical portion of the submitted work lacks sufficient details regarding the experimental procedures, making it difficult to assess the results. Moreover, discrepancies between the data presented in the text and the figures are apparent.
Based on similar publications (doi: 10.1039/c9cp00548j; doi.org/10.1016/j.electacta.2022.141058), the authors should provide more comprehensive data on the electrochemical measurements. In particular, the electrochemical impedance spectroscopy (EIS) experiments lack essential information about the experimental setup. For RFBs, parameters such as the state of charge, electrode geometry, electrode spacing, and the potential at which the measurement is performed significantly influence the obtained results. The absence of these details suggests that the authors may not have a complete understanding of the research topic.
Furthermore, the proposed equivalent circuit does not align with the assumed model, which considers two dominant phenomena: charge transfer resistance of the electrochemical reaction and diffusion processes. The proposed circuit does not conduct faradaic current, preventing electrochemical reactions from occurring, and lacks an element representing the diffusion process. Additionally, it seems unlikely that the battery performance would change so dramatically with the alteration of the kinetics of only one of the two electrode reactions, especially considering that the charge transfer resistance of this reaction is smaller. The overall rate is typically determined by the slowest process, which remains largely unaffected.
Figures 4c and 4d present parameters without a clear explanation of their determination method, and the reported numerical values appear implausible. The values in Figure 4c are excessively large, while those in Figure 4d are considerably too small.
In the manuscript introduction, the detrimental processes occurring in the battery should be explained in more detail, with appropriate literature references
Reviewer 3 Report
Comments and Suggestions for Authors
This work by Huishan Chen et al. reports on carbon fiber decoration with Bi for use as electrode in vanadium flow batteries (VFB). The authors explain that Bi particles has already been shown promising in this role elsewhere and here they suggest a better method of decoration, namely, hydrothermal. More detail would be expected about this method: I wonder how, technically, treatment in an aqueous solution at 180°C was carried out: under pressure?
Then, I found no evidence on mass loading of Bi, which is essential.
Concerning the surface of treated/untreated fibers, the text reads "specific surface area of Bi-GF is 5.862 m2/g, almost ten times that of 0.624 m2/g for GF " referring to Table S1. But Table S1 gives different values, without units:
Table S1. BET specific surface areas of different electrodes.
electrode Specific surface area
GF 4.81
Bi-GF 5.862
Which is correct?
Supposing the text is correct in claiming 1-fold increase in the electrode surface, in accord with the impression from microscopy photos, the issue of current, charge, and resistance normalization per surface area becomes crucial: what do cm2 in all the graphs and text imply? If it is just geometric area, then about 10-fold changes could be expected after Bi deposition just due to the available surface increase. Correct characterization would require determination of electrochemically available surface area (EASA) via capacitive currents in an inert electrolyte and normalization by this EASA. In this way, specific effects due to Bi would be separated from mere electrode surface increase.
Round 2
Reviewer 1 Report
Comments and Suggestions for Authors
No further comments. The authors responded to all previous questions.
Author Response
We thank the reviewer for their acceptance of our revised manuscript.
Reviewer 2 Report
Comments and Suggestions for Authors
The manuscript has been revised by the authors, unfortunately they did not address all my comments, such as the one regarding the determination of the electrolyte's state of charge used in impedance measurements or the fact that the rate of a complex reaction depends on the rate of the slowest step, which did not change in the system with a modified electrode. Additionally, the method of calculating the potential losses and overpotential presented in Figures 4c and 4d remains a mystery. In this case, the authors only changed the values on one of the graphs (currently the numerical values are 1000 times smaller). Below I present comments mainly on impedance measurements:
- Mixing two different solutions will result in a solution with concentrations of individual species smaller than the initial solutions - in the case of the presented experiment, this law does not work.
- The authors provide the value of the potential to which the studied electrodes are polarized during EIS measurements, unfortunately they do not specify with respect to which electrode this value refers and what is the value of the open-circuit potential (equilibrium), because it is not known whether the studied electrode is polarized in the cathodic or anodic direction and by what value relative to the equilibrium value.
- It should be borne in mind that published works are not free from errors and the role of the reviewer is, among others, to prevent the replication of these errors. In the case of the equivalent circuit stubbornly used by the authors, I cannot accept it. Firstly, the diffusion process cannot be represented by a capacitor, as it does not allow for the flow of Faraday current, i.e., it does not allow the cell to work. Moreover, it is widely known (without the use of an equivalent circuit) that the diameter of the semicircle visible on the Nyquist plot corresponds to the charge transfer resistance in the electrode reaction - such an interpretation can be found in practically every work on this subject. In the case of the reviewed manuscript, the value of the resistance R(ct) clearly differs from the value of the diameter of the obtained semicircles. I suggest following the article (https://doi.org/10.1016/j.mex.2019.03.007), the authors of which have extensive knowledge and experience in EIS measurements in flow batteries - the equivalent circuit they use for individual half-cells is the simplest correct interpretation of electrode processes.
- The charge transfer resistance was determined for only one electrode reaction, which is not mentioned in the comment on this part of the study, which may suggest that the overall reaction rate is being analyzed. There is no information about the work of the second electrode and the analysis of the spectrum for this (negative) electrode - looking at the spectra, the change is negligibly small. Here, too, my comment on the overall reaction rate. In the case of complex reactions, the overall reaction rate is determined by the slowest process. Since the rate of the slowest process has not changed, where does such a spectacular change in the parameters of the entire cell come from?
- Unfortunately, the lack of description of the experiment and the parameters inconsistent with the presented results also apply to section 2.5. There is no information about the amount of electrolyte used, and thus there is no information about the state of charge of the electrolyte used during cell operation. Additionally, the authors write that the battery tests are carried out in the voltage range from 0.7V to 1.2V, which does not agree with the graphs presented in the work. Moreover, in Figures 4a and 5a, the charging and discharging curves are presented for a current density of 200mA/cm2. These curves intersect, which is impossible. This is already a very high value of current density and the voltage range during charging is different from the voltage range during cell discharge (of course, taking into account the same working current).
Reviewer 3 Report
Comments and Suggestions for Authors
I am quite satisfied by the author's response and have no more objections for the paper to be published.
Author Response

(The authors gave the same response as above.)

Round 3
Reviewer 2 Report
Comments and Suggestions for Authors
The authors have made corrections following my second review. The new information allowed me to verify another part of the experimental data. Unfortunately, I disagree with the value of the charge delivered or withdrawn from the battery during the tests presented in Figure 5a. The 0.1 M vanadium salt solution used in the experiment, in an amount of 20 ml (0.02 dm³), requires a charge of Q = 0.1 * 0.02 * 96500 = 193 C = 0.054 Ah to change the oxidation state of the total amount of vanadium ions by one, while the battery tests were conducted by supplying or withdrawing a charge up to a value of 0.8 Ah, which is many times more than is theoretically possible.
The determination of overpotential has not been explained. The theoretical definition is easy to give, but performing calculations for an operating system is a different matter and much more difficult. The authors have indeed included a literature reference on this issue (number 21), but unfortunately it does not mention the calculation of overpotential or equilibrium potential. I have a few questions here: 1) The authors calculate the overpotential during battery tests, stating that they calculate it for the electrode, so how do they measure the potential of a single electrode in an operating cell? 2) To calculate the electrode potential at equilibrium for concentrated solutions (with which they work), we need activities rather than their concentrations. Even if concentrations were used, which ones? Averaged for the entire solution or averaged for the electrode space? If the latter, how to calculate them? The estimation of concentrations or activities is subject to a large error, which should be calculated and taken into account in further considerations.
It is also surprising that the authors perform battery tests using such a dilute solution of vanadium salt and for such high current densities. Such tests are performed on solutions with concentrations from 1.5 M to 2 M, i.e., 15 to 20 times more concentrated (doi.org/10.1021/acsaem.2c03891; doi.org/10.1063/1.4800202; DOI: 10.1039/c7cp02581e).
I was very surprised by the change in the Nyquist plot (Figure 3d) for the negative electrode? Where do such changes come from compared to the previous version? On the current graph, the electrode reaction occurring on the negative electrode is not visible and the range of values on the z-axis with the real value of the measured impedance is much smaller than on the graph in the previous version of the manuscript.
The results seem unreliable, there are too many inconsistencies and significant unexplained changes.
Round 4
Reviewer 2 Report
Comments and Suggestions for Authors
The authors have made further revisions to the manuscript. They have removed some of the results as they were not reliable. Following my last comments regarding the electrolyte concentration, the authors have stated that they made a mistake and the concentration is ten times higher than previously declared. This new electrolyte concentration does not allow for the declared capacity in Figure 4a. An electrolyte with a concentration of 1M vanadium salt and a volume of 20 ml can theoretically accept or release a charge of Q=1M*0.02dm3*96500C/mol = 1930 C, which corresponds to 0.536 Ah, which is less than the value of 0.8 Ah shown in Figure 4a for both the charging and discharging process and declared in the summary of the article. In the case of impedance measurements, the results should be shown over the entire frequency range and additionally, a magnification of some fragment if it is justified. In the latest version of the manuscript, the authors have only shown part of the Nyquist plot for negative electrode, so this needs to be corrected.
